# Elastic-Plastic Threshold and Rational Unloading Level of Rocks

**Ming Ji [1,*] and Hongjun Guo [1,2]**

1. Key Laboratory of Deep Coal Resource Mining, Ministry of Education of China,
   China University of Mining & Technology, Xuzhou 221116, China
2. School of Construction Management, Jiangsu Vocational Institute of Architectural Technology,
   Xuzhou 221116, China
* Correspondence: jiming@cumt.edu.cn

**Abstract:** During loading and unloading test, various rocks manifest different stress values of elastic-plastic transformation. This study proposes to include axial pressure increment ratio in the conventional triaxial compression test to evaluate different variables (nominal elastic modulus, nominal Poisson's ratio, strain, and energy). The relationships among various factors including variables, the stress level of initial confining stress and axial pressures, were analyzed by analyzing the stress–strain plot record obtained from testing various rocks. The extreme value point of the deformation parameter, also known as the elastic-plastic threshold, was analyzed. In addition, the elastic-plastic thresholds were later used as unloading points during the unloading tests. Under the same confining condition, different rocks demonstrated different unloading levels. Furthermore, a linear correlation was observed between unloading levels and changing confining pressures, and the gradient is mainly related to the types of rocks. During the unloading tests of rocks, the rational unloading level is recommended to be no higher than the stress level at the elastic-plastic threshold under the corresponding confining pressure.

**Keywords:** triaxial compression test; unloading level; deformation parameter–axial pressure increment ratio; strain–axial pressure increment ratio; energy–axial pressure increment ratio; elastic-plastic threshold

## 1. Introduction

Rock mechanics studies have been extensively conducted in China and abroad, which cover a broad range, from uniaxial to triaxial compression, simple to multiple conditions, and single to complex paths [1–6]. In triaxial tests, the initial stress level mainly includes the initial confining pressure and the initial axial pressure. Apart from the properties of rocks, the axial pressure is mainly affected by the confining pressure and the intrinsic attributes of rocks. During a triaxial unloading test, the initial axial pressure (i.e., unloading axial pressure or unloading level) is typically lower than the peak strength under the corresponding confining pressure. According to the results obtained from conventional triaxial compression tests, Liu [7] and Li [8] determined that 85–95% of the peak strength is the unloading point under the corresponding confining pressure. Liu et al. [9] selected different levels of initial axial pressure (70%, 80%, and 90%) for unloading tests on the sandstone. The research revealed that decreasing the initial axial pressure could cause lower ultimate strength and axial strain of rocks during unloading accompanied with more pronounced brittleness and expansion characteristics. Zhang et al. [10] investigated the relationships between elastic strain energy and axial strain while the unloading stresses were set at levels of 60% of the peak stress and 80% of the peak stress. The results indicated that the energy change curves at both levels are parallel after unloading the confining pressure

and the unloading level has no significant effect on the energy evolution process. Zhang [11] and Zhu [12] considered that the unloading stress level mitigated possible damage within rocks during the unloading damage stage. Regarding the initial axial pressure of unloading, Liu et al. [13] selected 50% of the conventional triaxial compressive strength and the corresponding confining pressure, whereas other researchers often adopted 60% and 80% [14–16], 70% [17–20], 80–90% [21], and 15–90% [22]. Most researchers selected 80% of the peak strength as the loaded axial pressure in unloading tests [23–26]. Details of studies discussed above can be viewed in Table 1. However, mentioned studies failed to consider the effects of the initial axial pressure during exploring the effects of unloading on rock failures.

**Table 1.** Threshold value of elastic-plastic transformation of rocks.

| Rock | Threshold Value of Elastic-Plastic Transformation | Researcher and References |
| --- | --- | --- |
| Sandstone | 85–95% | Liu [7] |
| Siltstone | 85–95% | Li [8] |
| Sandstone | 70%, 80%, 90% | Liu [9] |
| Marble | 60% and 80% | Zhang [10] |
| Gneiss | 55–65% | Zhang [11] |
| Shale | About 65% | Zhu [12] |
| Marble | 50% | Liu [13] |
| Marble | 60% | Cong [14,15] |
| Mudstone | 80% | Deng [16] |
| Marble | 70% | Chen [17,18], Zhao [20] |
| Schist | 70% | Yang [19] |
| Rhyolite | 80–90% | Zhong [21] |
| Siltstone | 15–90% | Ji [22] |
| Granite, red sandstone | 80% | Du [23] |
| Granitic rock | 80% | Dai [24] |
| Sandstone | 80% | Qin [25] |
| Marble | 80% | Zhao [26] |

At the elastic deformation stage, the deformation of rocks caused by external load is reversible and recoverable, which produces little effects on the rocks per se. Nonetheless, at the plastic deformation stage, the loading force causes the developments of microcracks in the rock, and stress concentration points at both ends of microcracks. If the load continues to increase, randomly distributed microcracks will transform into regular penetrating macrocracks, resulting in irreversible deformation and damage. During other unloading tests under the condition of relatively large axial pressure level (plastic deformation stage), the existing damage of the test specimens may cause errors in testing result and therefore can no longer be ignored. Tests suggest that selecting a rational initial axial pressure for unloading tests is critical.

This study combines the conventional triaxial compression test with axial pressure increment ratio to find out the critical inflection points by monitoring compressive deformations of the rock under different confining pressures [27]. Conducting unloading tests of rocks under complex stress paths provides a theoretical reference for the selection of initial axial pressure.

## 2. Mechanical Study of Siltstone Under Loading

### 2.1. Test Protocol

Rock samples were processed into cylindrical standard specimens (with a diameter of 50 mm and a height of 100 mm as shown in Figure 1). The specimens, which demonstrated integrity and uniformity, and little dispersion of wave velocity in wave velocity tests, were selected for loading tests with different confining pressures. The tests were performed with the MTS 815.02 Electro-hydraulic Servo-controlled Rock Mechanics Testing System (MTS Systems Corporation, Eden Prairie, MN, USA) at the China University of Mining and Technology (Xuhzou, Jiangsu, China; Figure 2).

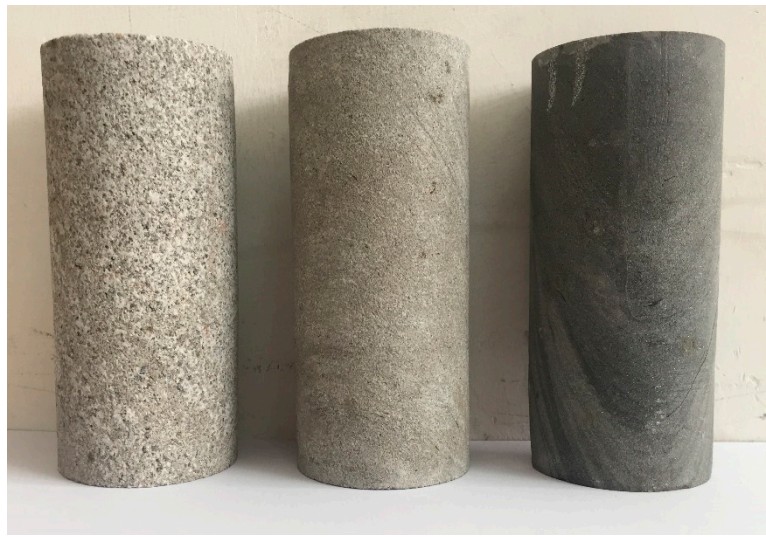

**Figure 1.** Typical rock specimens.

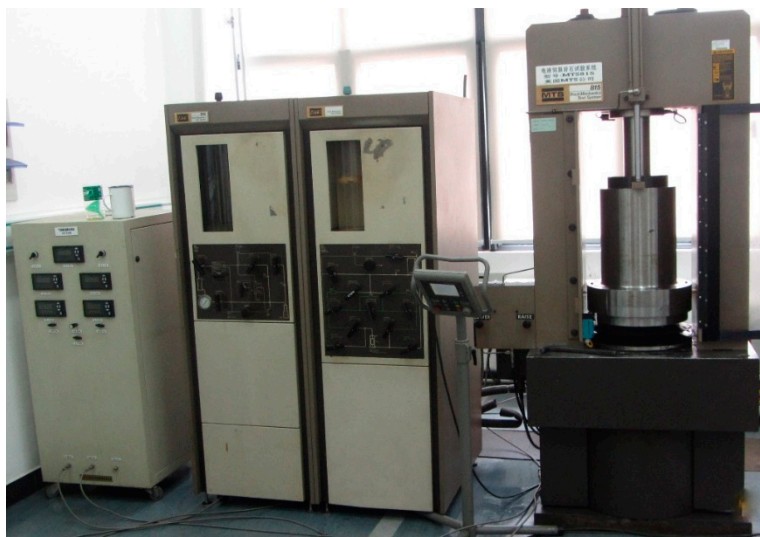

**Figure 2.** MTS 815.02 Electro-hydraulic servo-controlled rock mechanics testing system.

The stress path and loading method used in the tests are shown in Figure 3. The specific procedures was described as follows: (1) alternating stress application, which refers to alternately apply confining pressures ($\sigma_2 = \sigma_3$) and axial pressures ($\sigma_1$) at a loading rate of 0.05 MPa/s to reach a preset confining pressure; (2) fixed stress application, which means a fixed confining pressure and an axial pressure were maintained at a loading rate of 0.25 MPa/s until failure of rock specimens; and (3) displacement pressure application, which means that the loading forces continued to apply after the peak to obtain the entire stress–strain curve.

### 2.2. Evolution Analysis of Deformation Parameter

This research focused on the stress state of rock specimens during the loading process under various hydrostatic pressures. Based on the test data, Figure 4 depicts the stress–strain curves of siltstone specimens in the conventional triaxial compression test at a confining pressure of 10 MPa, wherein $\varepsilon_1(\varepsilon_3, \varepsilon_v)$ is the axial (lateral, volumetric) strain of siltstone specimens which occurred under the load.

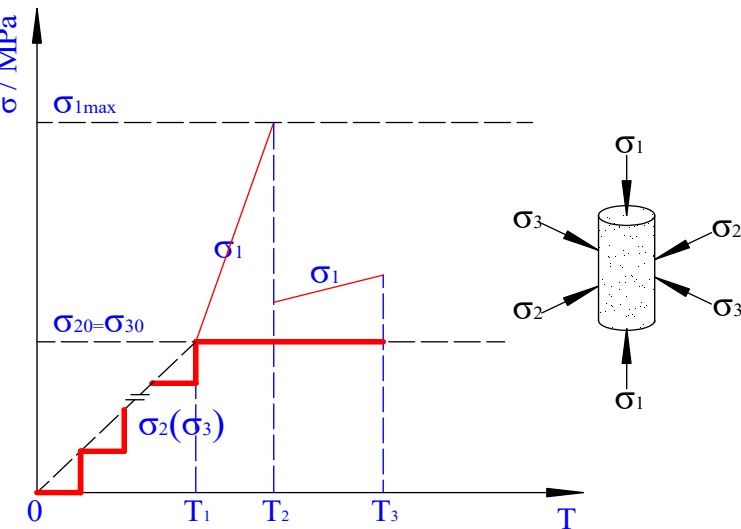

**Figure 3.** Stress path and loading method used in the conventional triaxial compression test.

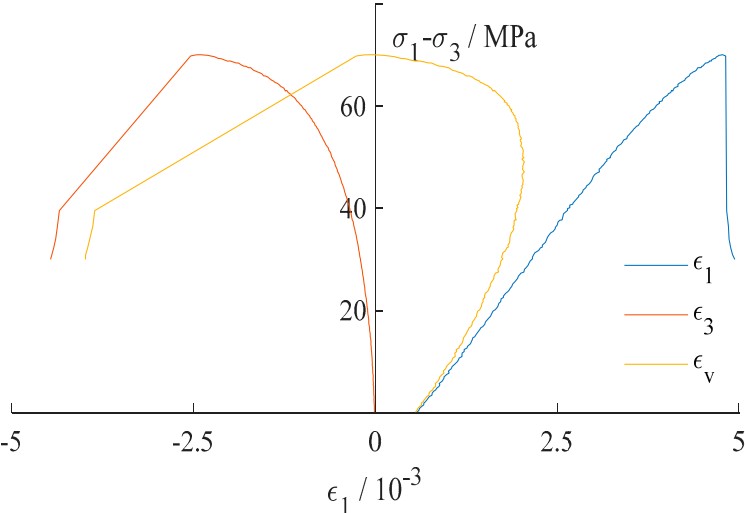

**Figure 4.** Stress–strain curves of siltstone specimens at confining pressure of 10 MPa.

Elastic modulus and Poisson's ratio are important mechanical parameters of rocks, whose changes are closely associated with rock deformation damage and the elastic-plastic transition. Assuming that rock specimens under three-dimensional stress follow the generalized Hooke's law, then they are consistent with the condition listed below:

$$\begin{cases} \varepsilon_1 = \frac{1}{\overline{E}}[\sigma_1 - \overline{\mu}(\sigma_2 + \sigma_3)] \\ \varepsilon_2 = \frac{1}{\overline{E}}[\sigma_2 - \overline{\mu}(\sigma_1 + \sigma_3)] \\ \varepsilon_3 = \frac{1}{\overline{E}}[\sigma_3 - \overline{\mu}(\sigma_1 + \sigma_2)] \end{cases} \tag{1}$$

where $\varepsilon_1(\varepsilon_2, \varepsilon_3)$ is the axial strain (lateral strain), $\sigma_1(\sigma_2, \sigma_3)$ is the axial pressure (confining pressure), $\overline{E}$ is nominal elastic modulus, and $\overline{\mu}$ is nominal Poisson's ratio.



Under the conventional triaxial compression, $\sigma_2 = \sigma_3$, Equations of nominal elastic modulus $\overline{E}$ and nominal Poisson's ratio $\overline{\mu}$ for the deformation parameter of rocks [28,29] are formulated below by substituting $\sigma_2 = \sigma_3$, into Equation (1), that is:

$$\begin{cases} \overline{E} = \dfrac{\sigma_1 - 2\overline{\mu}\sigma_3}{\varepsilon_1} \\ \overline{\mu} = \dfrac{\frac{\varepsilon_3}{\varepsilon_1}\sigma_1 - \sigma_3}{\left(2\frac{\varepsilon_3}{\varepsilon_1} - 1\right)\sigma_3 - \sigma_1} \end{cases} \tag{2}$$

Under a fixed confining pressure, the change in deformation parameter is only related to axial pressure. To analyze the effects of axial pressure change, we introduce the concept of deformation parameter–axial pressure increment ratio (Qiu et al. 2012), namely the ratio of nominal elastic modulus (nominal Poisson's ratio) increment to the axial pressure increment, which is expressed as:

$$\begin{cases} \dfrac{\Delta\overline{E_{t+1}}}{\Delta\sigma_{1(t+1)}} = \dfrac{\overline{E_{t+1}} - \overline{E_t}}{\sigma_{1(t+1)} - \sigma_{1(t)}} \\ \dfrac{\Delta\overline{\mu_{t+1}}}{\Delta\sigma_{1(t+1)}} = \dfrac{\overline{\mu_{t+1}} - \overline{\mu_t}}{\sigma_{1(t+1)} - \sigma_{1(t)}} \end{cases} \tag{3}$$

where $\Delta\overline{E_{t+1}}$ is the increment of nominal elastic modulus at time $(t+1)$, $\overline{E_{t+1}}, \overline{E_t}$ is nominal elastic modulus at time $(t+1)$ (and $t$), $\Delta\overline{\mu_{t+1}}$ is the increment of nominal Poisson's ratio at time $(t+1)$, $\overline{\mu_{t+1}}, \overline{\mu_t}$ is nominal Poisson's ratio at time $(t+1)$ (and $t$), $\Delta\sigma_{1(t+1)}$ is the increment of axial pressure at $(t+1)$, and $\sigma_{1(t+1)}\left(\sigma_{1(t)}\right)$ is the axial pressure at time $(t+1)$ (and $t$).

The strain–axial pressure increment ratio is to describe a physical quantity, indicating the change rate of strain as a function of axial pressure. This ratio can sufficiently reflect the effects of axial pressure change on the deformation damage of rocks during loading, as well as the response rate of the internal structure to the external macro mechanical state change. The evolution of the deformation parameter during the complete stress–strain process of siltstone and the curve between the axial pressure increment ratio and the stress level are shown in Figure 5.

As the axial pressure increases, some irreversible deformation gradually occurs within the rock. Continuous pressure loading leads to rock degradation and an accumulation of damage until failure. From the hydrostatic pressure state to failure, several successive stages are observed including linear–plastic–failure–residual. During these stages, the deformation parameter undergoes the following process: "rapid strengthening, low-speed strengthening, low-speed weakening, drastic drop, and residual strain." This finding is consistent with the conclusion obtained from Figure 5. The nominal Poisson's ratio changes in a U-shaped pattern (first decreasing and then increasing) with increasing axial pressure. Conversely, the nominal elastic modulus exhibits an inverted U-shaped pattern (first increasing and then decreasing) with increasing axial pressure.

Therefore, we selected the extreme value of the deformation parameter as the elastic-plastic threshold during loading of rocks. Prior to loading at this threshold, the rock is intact with no detected interior damage. Afterwards, varying degrees of damage and failures have been found within rocks. The nominal elastic modulus inflected (from increasing to decreasing) at the 61.17% stress level, and its axial pressure increment ratio, which is sensitive to axial pressure change, showed a pronounced change at the 60.17% stress level. The nominal Poisson's ratio also inflected (from decreasing to increasing) at the 61.17% stress level, but its axial pressure increment ratio presents no pronounced change at stress levels below 86%. Therefore, both the extreme value of nominal elastic modulus (nominal Poisson's ratio) and the stress levels corresponding to the first pronounced change of its axial pressure increment ratio can be taken as the sign of elastic-plastic transition and damage (or not) of rocks.

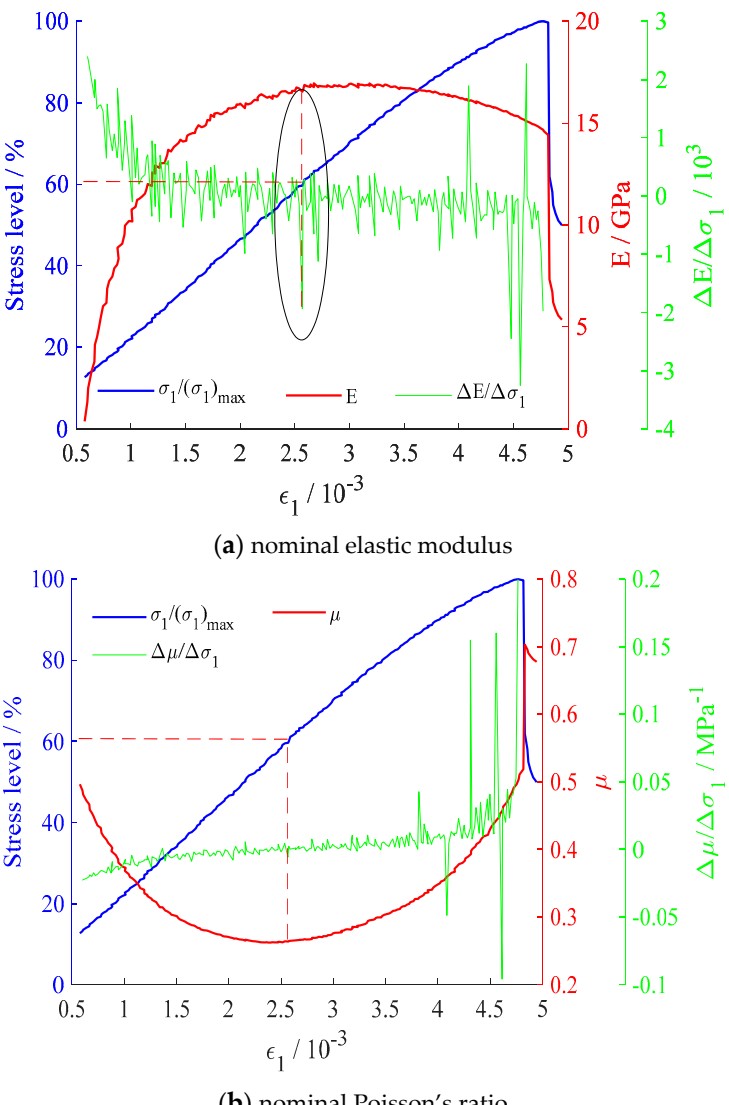

(**a**) nominal elastic modulus

(**b**) nominal Poisson's ratio

**Figure 5.** Evolution curve of deformation parameter–axial pressure increment ratio.

### 2.3. Evolution Analysis of Strain

Based on the concept of axial pressure increment ratio, the strain–axial pressure increment ratio is calculated using the above-mentioned method, namely:

$$\frac{\Delta\varepsilon_{i(t+1)}}{\Delta\sigma_{1(t+1)}} = \frac{\varepsilon_{i(t+1)} - \varepsilon_{i(t)}}{\sigma_{1(t+1)} - \sigma_{1(t)}} \tag{4}$$

where $\Delta\varepsilon_{i(t+1)}(i = 1, 3, v)$ is the axial (transverse, volumetric) strain increment at time $(t + 1)$ and $\varepsilon_{i(t+1)}\left(\varepsilon_{i(t)}\right)$ is the strain at time $(t + 1)$ (and $t$).

A greater strain–axial pressure increment ratio of a particular direction indicates that the deformation of this direction is more sensitive to axial pressure change. We combined the test data and Equation (4) to solve the relationship between the strain–axial pressure increment ratio and the stress level of siltstone specimens (Figure 6).

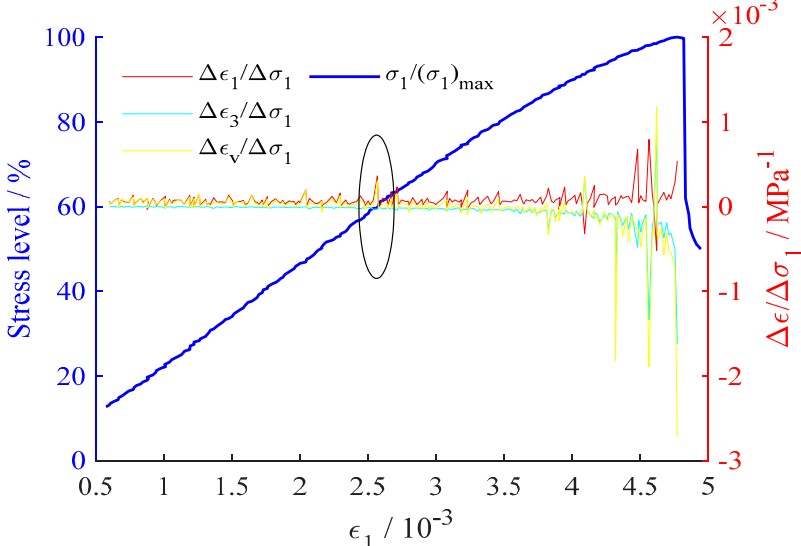

**Figure 6.** Evolution curve of strain–axial pressure increment ratio.

The overall change of the strain–axial pressure increment ratio was not pronounced (Figure 6). Nonetheless, according to the stress level at which the elastic-plastic threshold appeared, evidently the first relatively pronounced change occurred at the 60.17% stress level. This result further confirms the rationality of the elastic-plastic threshold we have selected.

### 2.4. Evolution Analysis of Strain Energy

According to the law of conservation of energy, the entire deformation failure process of a rock is associated with energy conversion. Following the law of thermodynamics, energy production is mainly derived from external forces. In the triaxial test, external forces imposed on rock specimens include the axial pressure and confining pressure of the testing equipment. Imposed axial pressure causes axial deformation of rock specimens, while the confining pressure leads to lateral deformation [26,30]. Therefore, the total energy $U$ absorbed by rock specimens includes the axial strain energy $U_1$ absorbed by axial deformation and the hoop strain energy $U_3$ consumed by lateral deformation. Then there is:

$$U = U_1 + U_3 \tag{5}$$

At any time $t$ during the test, the axial strain energy $U_1$ and the hoop strain energy $U_3$ can be derived from the stress–strain curve integral, that is:

$$U_1 = \int_0^{\varepsilon_{1(t)}} \sigma_1 d\varepsilon_1 \tag{6}$$

$$U_3 = 2\int_0^{\varepsilon_{3(t)}} \sigma_3 d\varepsilon_3 \tag{7}$$

where $\varepsilon_{1(t)}\left(\varepsilon_{3(t)}\right)$ is the axial (transverse) strain at any time $t$.

According to the concept of definite integral in Equations (6) and (7), the infinitesimal method is adopted to obtain the sum of the area, that is:

$$U_1 = \sum_{t=0}^{t_0} \frac{1}{2}\left(\sigma_{1(t+1)} + \sigma_{1(t)}\right)\left(\varepsilon_{1(t+1)} - \varepsilon_{1(t)}\right) \tag{8}$$

$$U_3 = \sum_{t=0}^{t_0} \left( \sigma_{3(t+1)} + \sigma_{3(t)} \right) \left( \varepsilon_{3(t+1)} - \varepsilon_{3(t)} \right) \tag{9}$$

Combined with the concept of axial pressure increment ratio, the strain energy–axial pressure increment ratio is expressed as:

$$\frac{\Delta U_{t+1}}{\Delta \sigma_{1(t+1)}} = \frac{U_{t+1} - U_t}{\sigma_{1(t+1)} - \sigma_{1(t)}} \tag{10}$$

where $\Delta U_{t+1}$ is the strain energy increment in the rock interior at time $(t+1)$ and $U_{t+1}(U_t)$ is the strain energy in rock interior at time $(t+1)$ (and $t$).

The evolution of strain energy during loading of siltstone specimens and the relationship between the axial pressure increment ratio and stress level are shown in Figure 7.

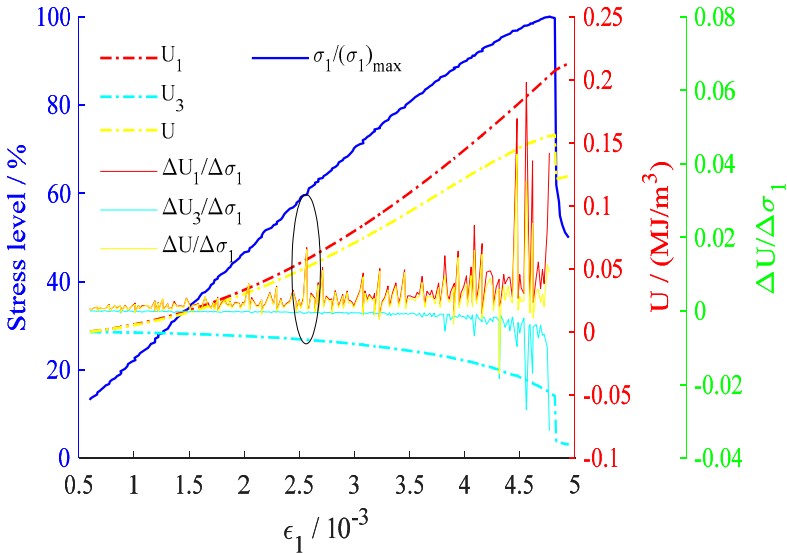

**Figure 7.** Evolution curve of strain energy–axial pressure increment ratio.

The example of axial strain energy (Figure 7) shows a pronounced increase in the rate of change at the 60.17% stress level. In other words, the strain energy–axial pressure increment ratio demonstrates the first pronounced change at this level, which is consistent with the stress level corresponding to the elastic-plastic threshold.

To sum up, a stress level at the elastic-plastic threshold during the loading process is observed. The parameters associated with the extreme deformation and the first pronounced response to axial pressure changes, together with the parameters for strain and strain energy, can be interpreted as the sign of the occurrence of this threshold.

## 3. Mechanical Analysis of Different Rocks Under Loading

To verify the conclusions described above, specimens with different lithology were selected for testing.

### 3.1. Mechanical Analysis of Mudstone Under Loading

Following the procedure described in Section 2 for the loading analysis of siltstone, changes in the mechanical properties of mudstone under loading at a confining pressure of 10 MPa are shown in Figure 8.

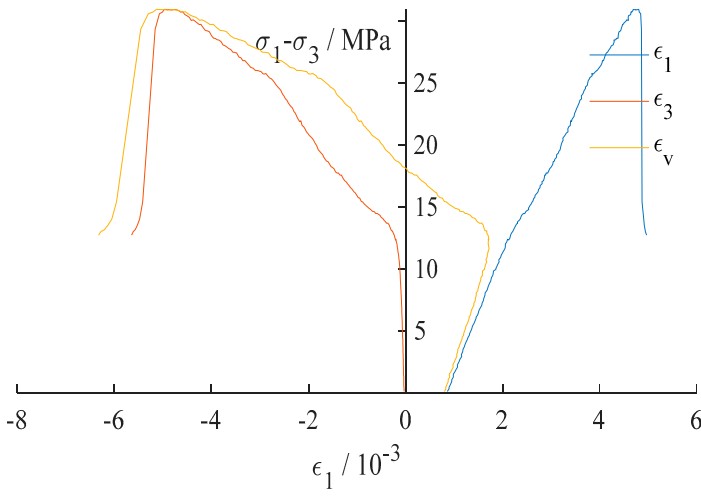

(**a**) Stress–strain

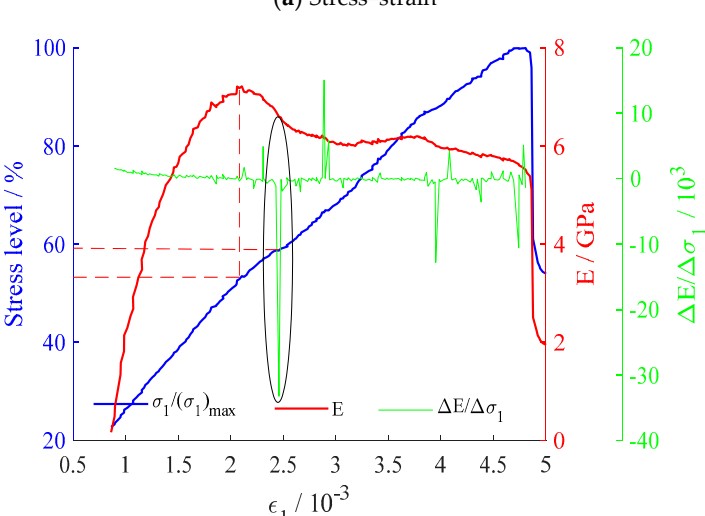

(**b**) nominal elastic modulus–axial pressure increment ratio

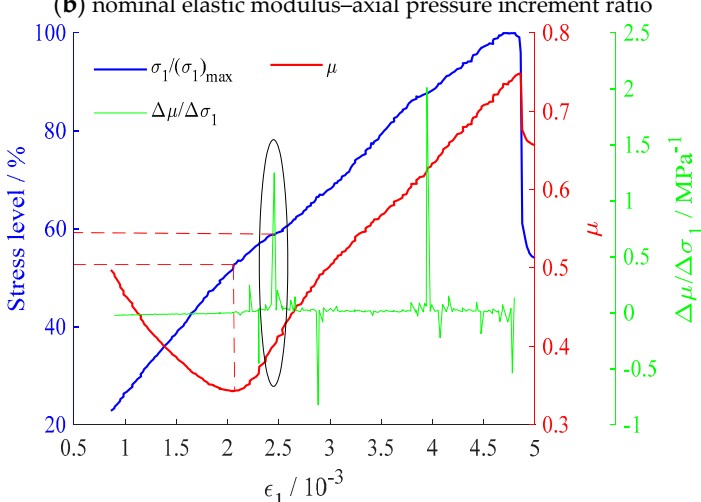

(**c**) nominal Poisson's ratio–axial pressure increment ratio

**Figure 8.** *Cont.*

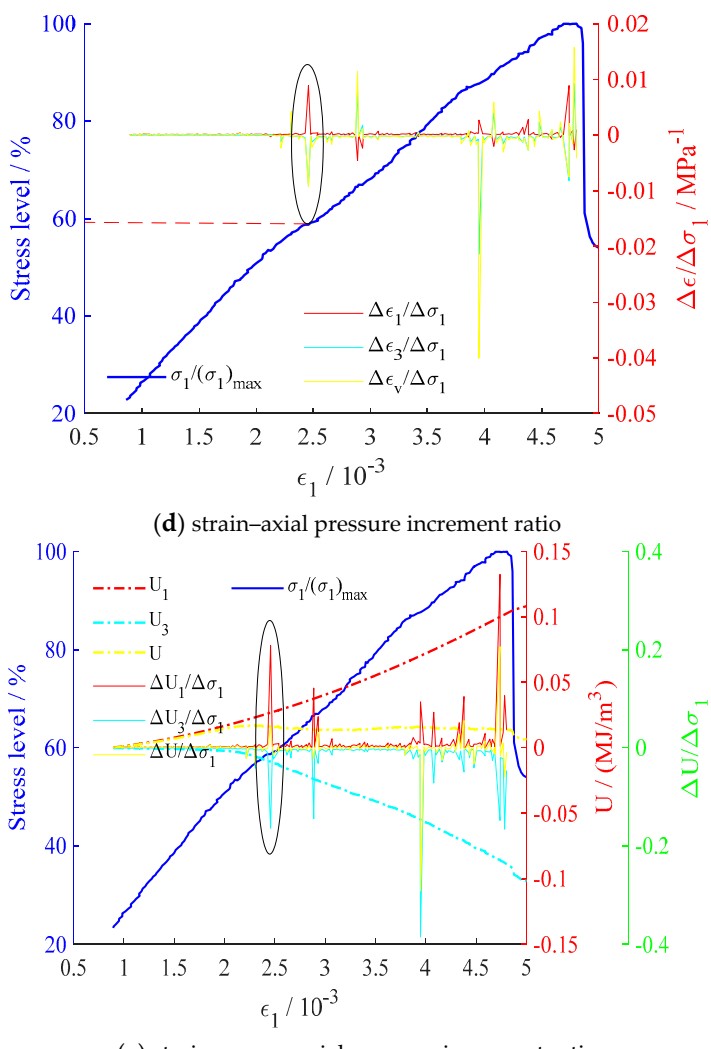

(**d**) strain–axial pressure increment ratio

(**e**) strain energy–axial pressure increment ratio

**Figure 8.** Evolution curve of mechanical properties of mudstone under loading.

Both the nominal elastic modulus and the nominal Poisson's ratio showed an inflection of increase or decrease at the 52.91% stress level (Figure 8). The axial pressure increment ratio of different variables shows a pronounced change for the first time at the 58.74% stress level, immediately followed by a significant change at the 66.51% stress level. The general trend is identical to what was observed from siltstone, and the obtained stress levels are similar, which lends further support to the conclusion. Meanwhile, the elastic-plastic threshold of various rocks differs during loading, which is associated with the intrinsic attributes and characteristics of rocks.

### 3.2. Exploration of Rational Unloading Level for Unloading Tests

When the loading tests and analysis results for siltstone, mudstone, and marble are combined, an elastic-plastic threshold in rocks during loading is revealed. At the corresponding stress level, the variables for rocks show an inflection of change or an initial sensitivity to axial pressure change. At this point, there is no irreversible deformation or failure inside the rocks. When the loading pressure continues, damage accumulates inside rocks. With increasing axial pressure, the scale of rock damage increases. During unloading of the confining pressure at higher stress levels, some damage has formed previously and cannot be eliminated, which has a definite impact on subsequent tests. Therefore, selecting a rational unloading level for unloading tests is vital.

To demonstrate the relationship between confining pressure and unloading level, the stress–strain curves of siltstone during loading at confining pressures of 20 and 30 MPa (Figure 9) are provided below.

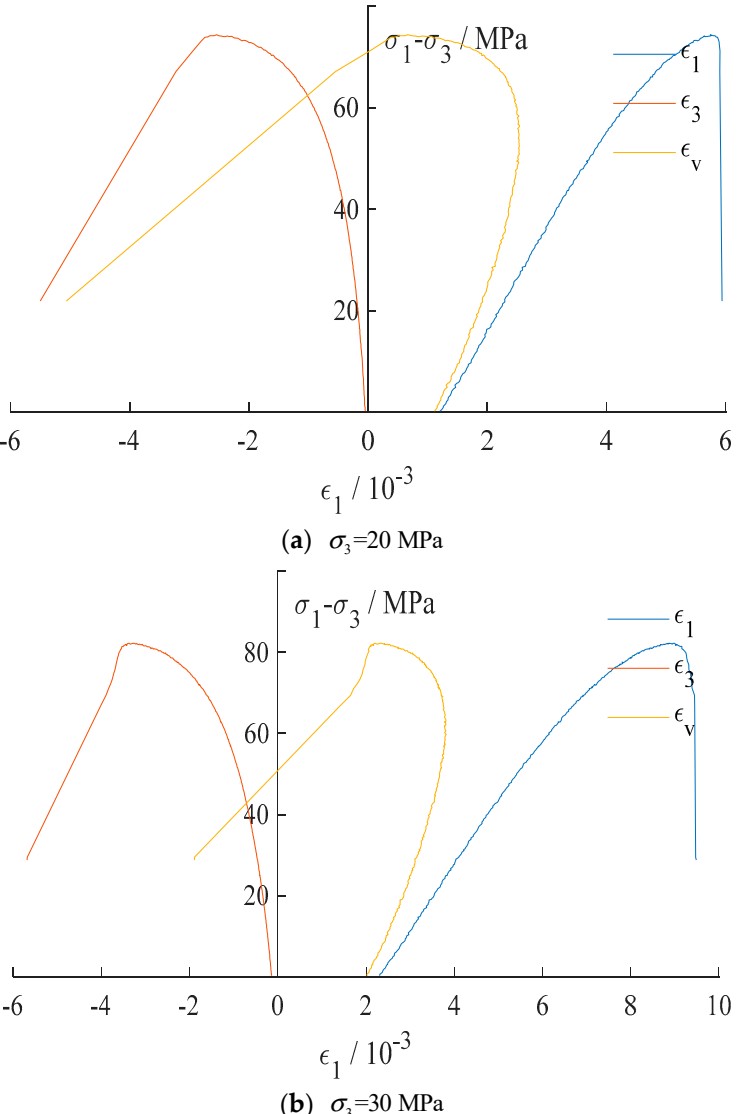

**(a)** $\sigma_3$=20 MPa

**(b)** $\sigma_3$=30 MPa

**Figure 9.** Stress–strain curves of siltstone under different confining pressures.

Taking the deformation parameter as an example, the relationships between the axial pressure increment ratio of nominal elastic modulus and nominal Poisson's ratio versus the stress level are shown in Figures 10 and 11, respectively.

According to in Figures 10 and 11, for siltstone, the stress level at the unloading point should be no higher than 67.47% at a confining pressure of 20 MPa, whereas at a confining pressure of 30 MPa, it should be no higher than 78%.

When the confining pressure is at 10 MPa, the stress level $\sigma_{ep}$, is expected to correspond to the threshold for an inflection in the deformation parameter, indicating whether a rock is damaged or not. The stress level $\sigma_{ep}$ exhibits an approximately linear relationship with the confining pressure (Figure 12). The fitted formula is expressed as

$$\sigma_{ep} = 0.9\sigma_3 + 50.3 \tag{11}$$

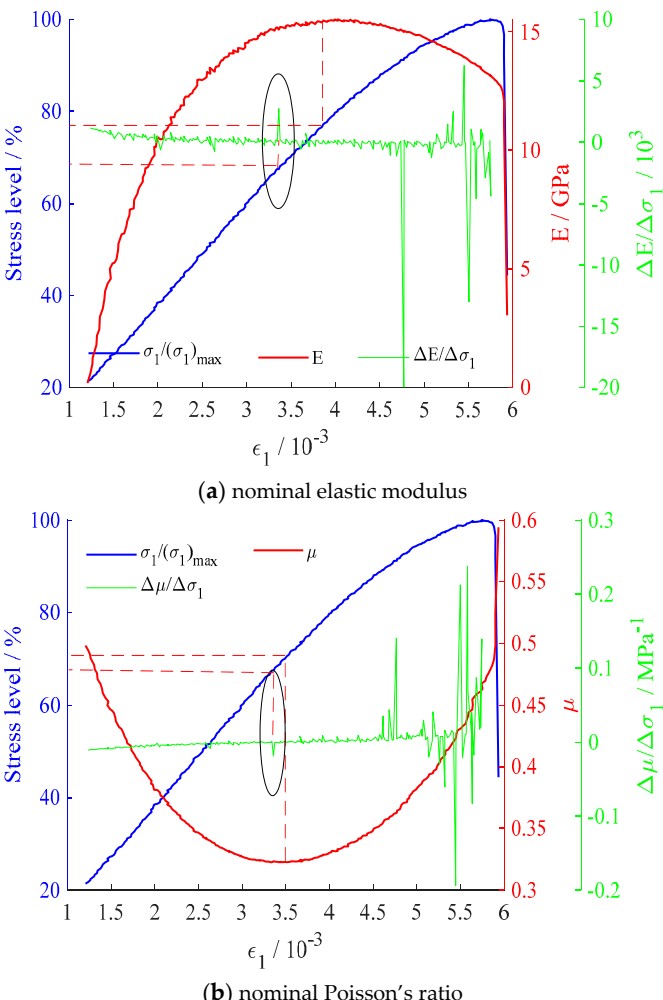

(**a**) nominal elastic modulus

(**b**) nominal Poisson's ratio

**Figure 10.** Evolution curve of deformation parameter–axial pressure increment ratio for siltstone ($\sigma_3 = 20$ MPa).

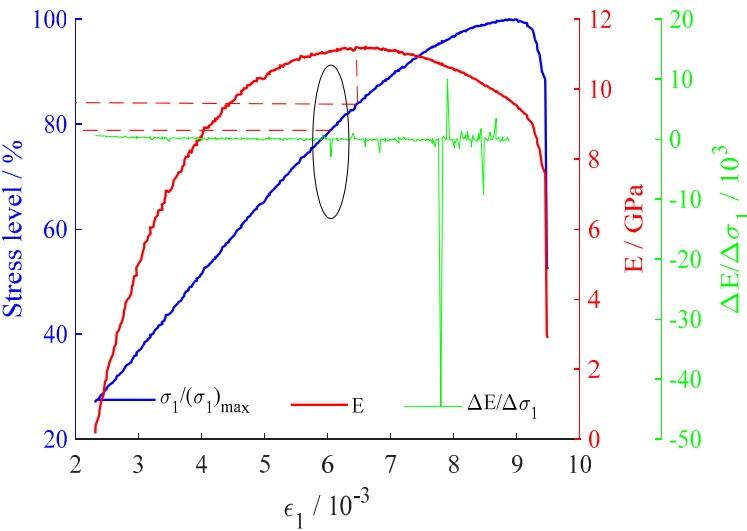

(**a**) nominal elastic modulus

**Figure 11.** *Cont.*

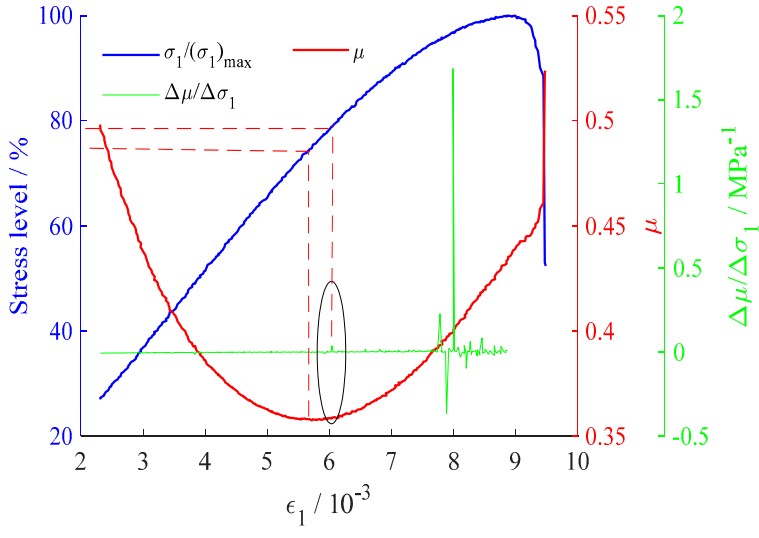

(**b**) nominal Poisson's ratio

**Figure 11.** Evolution curve of deformation parameter–axial pressure increment ratio for siltstone ($\sigma_3$= 30 MPa).

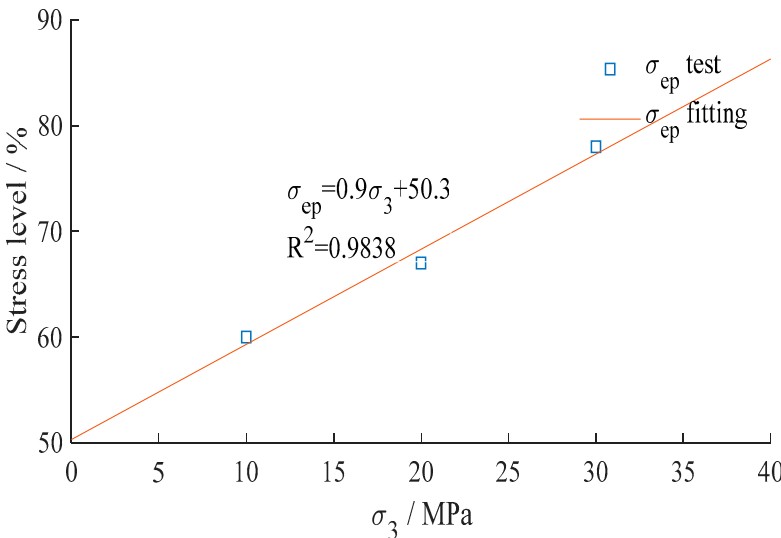

**Figure 12.** Relationship between unloading stress level and confining pressure.

It should be noted that the coefficient in the formula is related only to the intrinsic characteristics of the rocks.

The deformation of rocks in the elastic stage is reversible, and no substantial damage is detected. Therefore, for the studies of unloading tests of rocks under complex stress paths, we recommend selecting an unloading level that is equal to or slightly lower than the stress level at the elastic-plastic threshold (extreme value point of the deformation parameter) of rocks. Especially when unloading under low confining pressure, a stress level slightly below the value point of the deformation parameter can effectively prevent interior damage of rocks caused by a relatively high stress level before unloading at the conclusion of later tests.

## 4. Conclusions

(1) The nominal Poisson's ratio changes highlight a U-shaped pattern while increasing axial pressure, namely first decreasing and then increasing. In contrast, the change of elasticity modulus shows

an opposite trend. At a fixed confining pressure, the critical point of change in these trends corresponds to equal or similar stress levels;

(2)    At the same confining pressure condition, the stress levels corresponding to the first relatively pronounced or more pronounced change in the axial pressure increment ratio of various variables (strain, nominal elastic modulus, nominal Poisson's ratio, and energy) for the same rock are similar or equal. Differences exist among rocks due to their intrinsic attributes and characteristics;

(3)    The unloading stress level of siltstone rocks increases linearly as the confining pressure increases. However, the gradients of increasement are varied among rocks with different lithology due to rock's inherent properties;

(4)    For studies on unloading tests of rocks under complex stress paths, the unloading level is recommended to be equal to or slightly lower than the stress level at the elastic-plastic threshold of rocks under the corresponding confining pressure.

**Author Contributions:** J.M. conceived and designed the experiments; G.H. performed the experiments; J.M. and G.H. analyzed the data; J.M. and G.H. contributed reagents/materials/analysis tools; J.M. and G.H. wrote the paper.

**Funding:** This paper is supported by "Priority Academic Program Development of Jiangsu Higher Education Institutions", and "the Fundamental Research Funds for the Central Universities (2017XKQY044)".

**Conflicts of Interest:** The authors declare that there is no conflict regarding the publication of this paper.

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
