# Peer review of "Elastic-Plastic Threshold and Rational Unloading Level of Rocks"

_applsci, doi:10.3390/app9153164_

Round 1

Reviewer 1 Report

- what was the inflection point, mention it in the abstract ?

- Currently,  there is no unified conclusion

   * Be careful with such bold comments. otherwise, properly cite to support this statement.

- remove "testing" from lime 28.

- something to help with visual demonstration.. in the intro , display diagram so the elastic-plastic stages via diagram

- ensure that you figures are big enough to observe clearly.. the font seems small.

Author Response

Thank you very much for your suggestion of my paper.

The changes of English language were marked with red color,and the replies to the reviewers were green.

(1)  what was the inflection point, mention it in the abstract ?

Reply: “the inflection point” is a mistake in the paper,I have changed it to “extreme value point”.  

(2)- Currently,  there is no unified conclusion

   * Be careful with such bold comments. otherwise, properly cite to support this statement.

Reply:The statement in the abstract is indeed imprecise. What I want to express is that:

The stress levels value of elastic-plastic transformation of different rocks in loading and unloading tests are also different.”

I have changed it in the abstract.

(3)- remove "testing" from lime 28.

Reply: I have removed it in the line 28.

(4)- something to help with visual demonstration.. in the intro , display diagram so the elastic-plastic stages via diagram

Reply:Following your suggestion, I have sorted out the result of all references and put the details in Table 1.

5- ensure that you figures are big enough to observe clearly.. the font seems small.

Reply: I have changed all the picture in the paper.

Reviewer 2 Report

Triaxial testing results are key inputs to pillar design. Identifying point of conversion from elastic to plastic behavior varies with rock and the variance is particularly contrasted between evaporate minerals and non-evaporate minerals. This research should be continued for more rock types in order to potentially derive relationships.

Suggest that the conclusions be simplified and graphical support be offered for the 2-"U" shaped Poisson's ratio statements.

Author Response

Thank you very much for your suggestion of my paper.

The changes of English language were marked with red colorand the replies to the reviewers were green.

 (1)Triaxial testing results are key inputs to pillar design. Identifying point of conversion from elastic to plastic behavior varies with rock and the variance is particularly contrasted between evaporate minerals and non-evaporate minerals. This research should be continued for more rock types in order to potentially derive relationships.

Reply: Thanks for your suggestion, in this paper we take sandstone as an example in the triaxial testing. You give us a new experiment direction,we will do the experiments on rock samples of coal strata in the future,and study the elastic to plastic behavior of them. And the editor just gives me 4days to modify the paper, we have no enough time to do the experiments of coal and any other rock samples, and I think my work now can be published as a paper. But we will do that, and compare the differences between rock and coal, it will be my next work. Thank you again.

(2)Suggest that the conclusions be simplified and graphical support be offered for the 2-"U" shaped Poisson's ratio statements.

Reply:I have simplify the conclusions,and 2-"U" shaped Poisson's ratio statements and elastic modulus were shown in Fig.5,and were described in chapter 2.2.

Reviewer 3 Report

Rocks, unlike other solid (manufactured) solid materials, are heterogeneous/non-homogeneous, anisotropic and discontinuous bodies, due to the diversity of component minerals, bonding cements and micro-tectonics. Although, geologically, the rock class to which they belong may be identical, the same types of rock behave very different from a geo-mechanical point of view.This is the result of a huge variety of behavioral patterns of rocks subjected to various loadings and any study in this respect is a significant contribution to knowledge of the rocks, from a geo-mechanical point of view. If the behavior of the certain types of rocks is relatively easy to study in the elastic domain, where the transformation of the rocks is relatively reversible and somewhat common for several types of rocks, the study of elastic-plastic domain where the rocks suffer an irreversible transformations/deformations is very complex and is, therefore, very different.Moreover, in situ, the rocks under the influence of stresses, induced by underground or surface excavations, to which they are subjected, which manifest in the form of loadings and sometimes in the form of “loading-unloading” cycles (e.g.: the rocks nearby the mining faces and digging workings), have a very complex geo-mechanical behavior. Considering all of this, from the beginning, we can say that this article is a particular interest in the mining and civil engineering area.The laboratory tests to which the rocks are subjected demonstrate an original use of the triaxial testing of the rocks in highlighting that the stress levels of the elastic-plastic transformation of different rocks in the loading and unloading tests are very different.The results from the data of the laboratory tests have a high degree of originality and are very useful in the theoretical knowledge of the post-elastic behavior of the rock samples, tested in the laboratory and in the extrapolation of these results in the field realities. The behavior of the rocks subjected to laboratory tests is very well explained, and the conclusions are relevant.The article is very well written, both in terms of graphics and text. In view of all this, I consider this article to show a particular interest and deserve to be disseminated by publishing it into a prestigious scientific journal, such as Applied Sciences.

Author Response

Thank you for your identification and support of the research of my paper. According to your suggestion, I invited a native American researcher to revise the English language of the paper. The language changes have been marked in blue color. Because there are too many changes, I will not list them one by one.

Thanks again!

Best wishes!
